# Pharmacokinetic Comparison of Chitosan-Derived and Biofermentation-Derived Glucosamine in Nutritional Supplement for Bone Health

**DOI:** 10.3390/nu14153213

**Published:** 2022-08-05

**Authors:** Hee Eun Kang, Seung Jin Kim, Eun-ji Yeo, Jina Hong, Arun Rajgopal, Chun Hu, Mary A. Murray, Jennifer Dang, Eunmi Park

**Affiliations:** 1College of Pharmacy and Integrated Research Institute of Pharmaceutical Sciences, The Catholic University of Korea, Bucheon 420-745, Korea; 2Department of Food and Nutrition, Hannam University, Daejeon 306-791, Korea; 3Access Business Group International, LLC, 5600 Beach Blvd., Buena Park, CA 90621, USA

**Keywords:** glucosamine, pharmacokinetics, osteoarthritis, bone health, dietary factors

## Abstract

Glucosamine and chondroitin sulfate have been used as nutritional supplementation for joint tissues and osteoarthritis (OA). Biofermented glucosamine is of great interest in the supplement industry as an alternative source of glucosamine. The purpose of this study is to compare the pharmacokinetics of chitosan-derived glucosamine and biofermentation-derived glucosamine as nutritional supplementation. In a randomized, double-blind and cross-over study design, we recruited subjects of healthy men and women. The pharmacokinetics of glucosamine were examined after a single dose of glucosamine sulfate 2KCl (1500 mg) with two different sources of glucosamine (chitosan-derived glucosamine and biofermentation-derived glucosamine) to male and female subjects fitted with intravenous (iv) catheters for repeated blood sampling up to 8 h. According to plasma concentration–time curve of glucosamine after an oral administration of 1500 mg of glucosamine sulfate 2KCl, AUC_0__–8h_ and AUC_0–__∞_ values of glucosamine following oral administration of chitosan-derived and biofermentation-derived glucosamine formulations were within the bioequivalence criteria (90% CI of ratios are within 0.8–1.25). The mean Cmax ratios for these two formulations (90% CI of 0.892–1.342) did not meet bioequivalence criteria due to high within-subject variability. There were no statistically significant effects of sequence, period, origin of glucosamine on pharmacokinetic parameters of glucosamine such as AUC_0__–8h_, AUC_0–__∞_, Cmax. Our findings suggest that biofermentation-derived glucosamine could be a sustainable source of raw materials for glucosamine supplement.

## 1. Introduction

Glucosamine is a small molecule (molecular weight (MW) = 179.17) that occurs naturally as amino monosaccharide and a normal constituent of glycosaminoglycans in the cartilage matrix and synovial fluid [1]. Glucosamine supplementations have been shown to exert specific pharmacological effects in joint tissues and when taken orally are considered symptomatic slow acting drugs for osteoarthritis (SYSADOA). Several randomized clinical trials have provided evidence higher efficacy of glucosamine compared to placebo for osteoarthritis [2]. Dietary supplements containing glucosamine and chondroitin sulfate have been widely used for the management of knee pain in osteoarthritis (OA) [3]. A network meta-analysis study that included 54 studies covering 16,427 knee OA patients indicated significant effects of glucosamine plus chondroitin in pain relief and function improvement compared to the placebo group. In addition, this combination group was the only treatment option exhibiting clinically significant improvement from baseline pain and function. Furthermore, no significant difference was observed with respect to adverse effects [4]. Glucosamine containing supplements have a slow onset of response and provide long lasting pain relief and functional improvement in osteoarthritis [5]. Another study by Multicenter Osteoarthritis intervention trial with SYSADOA, the MOVES study, has shown that supplementation with glucosamine and chondroitin sulfate showed effectiveness in reducing pain, stiffness and functional effectiveness [6].

Glucosamine is generally derived from chitosan that are found in the shells of shrimp and other sea crustaceans. Chitosan (poly-D-glucosamine) is a polymer of glucosamine sugars, specifically glucosamine and N-acetylglucosamine [7]. Since the majority of glucosamine raw materials used in dietary supplements are derived from shellfish, there is a growing consumer perception that these supplements will cause acute allergic reaction to people with shellfish allergy [8]. Another concern among consumers is the sustainability of the manufacturing process of raw materials that go into a product; this has become a growing concern in the supplement industry [9]. Glucosamine derived from the shells of sea crustaceans is not a very sustainable process; hence, it has become imperative to use glucosamine derived from a more sustainable process. Recently biofermented glucosamine became available in the dietary supplement industry and is being used under the assumption that it will be as effective as chitosan glucosamine.

In this paper, we have compared the pharmacokinetic profiles of classic glucosamine ingredient sderived from chitosan and glucosamine derived by fermentation.

## 2. Materials and Methods

### 2.1. Study Design and Subjects

A double-blind, single dose, fasting, and randomized cross-over design with one week washout period was conducted. The sample size of subjects in this study was calculated using G*Power 3.0 [10]. When the significance criteria (α) 0.05, comparison group 2, medium effect size 0.50, and statistical power (1 − β) 0.80 were set, the required number of subjects was calculated to be 18. In consideration of the dropout rate of 10%, the final 20 subjects were selected.

Healthy male and female adults were recruited and assigned to take three capsules together per day of bio-fermented glucosamine sulfate 2KCL 1500 mg (*n* = 10, 5 male and 5 female) or Chitosan derived glucosamine sulfate 2KCL 1500 mg (*n* = 10, 5 male and 5 female) in the first period. Two subjects were reported irritation due to skin catheter for eight hours. One week after the wash-out period, subjects took three capsules of bio-fermented glucosamine sulfate 2KCL 1500 mg or Chitosan derived glucosamine sulfate 2KCL 1500 mg on the cross-over study. Two subjects who had skin irritation catheter were excused the second period of their own wishes. Blood samples (3 mL) for PK analysis were collected into EDTA tubes via intravenous catheter. Additionally, the blood samples were collected at regular time points (0, 10 min, 30 min, 1 h, 2 h, 3 h, 4 h, 6 h) up to 8 h for PK analysis according to a previous study [11]. Plasma samples were collected by centrifuge at 4 °C (1500× *g* for ten minutes) from anticoagulated whole blood. All the samples were stored at −80 °C until the blood samples were analyzed. The general information and blood biochemistry of subjects were collected, including age, gender, weight, height, body mass index (BMI), vital sign (body temperature, pulse, systolic blood pressure, diastolic blood pressure), hemoglobin, hematocrit, BUN, creatinine, AST, ALT, ALP, and total bilirubin. The study protocol was approved by the Hannam University review boards (IRB No 2019-01-05-1218).

### 2.2. Glucosamine Sample Preparation

Biofermented glucosamine and chitosan-derived glucosamine sulfated 2KCl capsules of identical in weight and final appearance were prepared at Access Business Group LLC. It is a cGMP (current Good Manufacturing Practice) nutritional supplement manufacturing site (5600 Beach Blvd, Buena Park, CA, USA). Both bio-fermented glucosamine sulfate and chitosan derived glucosamine sulfate are white free-flowing powders and bitter in taste. While chitosan derived glucosamine is from shellfish, bio-fermented glucosamine sulfate is produced via fermentation on non-GMO corn. It is USP grade, conforming to USP specifications. Each capsule contains solely 500 mg of glucosamine sulfate in a clear hypromellose hard-shell capsule with no other active or inactive ingredients. Three capsules would deliver 1500 mg glucosamine sulfate. The two products, biofermented glucosamine (Lot # 11823-87-JD) and chitosan-derived glucosamine capsules (Lot # 11823-95-JH), were encapsulated and packaged separately on different days to prevent any potential cross contamination.

### 2.3. LC-MS/MS Analysis of Glucosamine

Glucosamine concentrations in human plasma samples were determined using the LC-MS/MS method developed [12]. In brief, 200 μL of acetonitrile with 20 ng/mL of metoprolol (internal standard, IS) was added to each 50 μL of plasma sample. After each sample had been mixed and centrifuged (16,000× *g*, 5 min), the supernatant was collected in a LC vial. Finally, 6-μL aliquots were injected directly onto the HPLC column described below.

The LC-MS/MS system featured an Agilent 6460 triple quadrupole platform fitted to the Agilent 1260 LC system (Agilent, Waldbronn, Germany). Instrument control and data acquisition were performed using Agilent MassHunter Workstation software (ver. B. 04. 00). Chromatographic separation was performed on a Luna 3 μm CN 100 Å column (2.0 × 100 mm; Phenomenex Inc., Torrance, CA, USA). The flow rate of the mobile phase (10 mM ammonium formate in distilled water:acetonitrile (30:70 (*v*/*v*))) was 0.3 mL/min. Glucosamine has two anomeric forms [13]. Separation of the two anomers of glucosamine were essential for avoiding ion suppression caused by plasma matrix. The total run time was 8.3 min for each sample. The electrospray ionization source of the mass spectrometer was operated in the positive ion mode. The instrument parameters were as follows: gas temperature, 280 °C; sheath gas temperature, 380 °C; gas flow, 10 L/min; sheath gas flow, 11 L/min; nebulizer pressure, 25 psi; and capillary voltage, 3500 V. The fragmentor voltages were 150 V for glucosamine and 117 V for the IS. The collision energies of glucosamine and the IS were 13 and 20 eV, respectively. The precursor-to-product ion transitions for glucosamine and IS were m/z 180.0 ([M + H] +) → 72.0 and m/z 268.3 ([M + H] +) → 116.2, respectively. The retention times of the two glucosamine anomers were approximately 3 and 4 min, and that for the IS was 7.5 min. The calibration range of glucosamine in plasma samples was 0.1–10 μg/mL, and the lower limit of quantification (LLOQ) was 0.1 μg/mL. The analytical method was validated for its selectivity, linearity, sensitivity, precision and accuracy in accordance with the US FDA guidelines for the validation of bioanalytical methods [14]. The mean intra- and inter-day coefficients of variation of the analyses of quality control samples (0.1, 0.3, 1, 5, 9 μg/mL) were below 9.02%; the accuracies were 85.3–103%.

### 2.4. Pharmacokinetic and Statistical Analysis

The total area under the plasma concentration–time curve from time zero to last measured time t and to infinity (AUC_0__–t_ and AUC_0–__∞_) was calculated using the trapezoidal rule–extrapolation method [15] using non-compartmental analysis in PKanalix software (Lixoft, Antony, France). The peak plasma concentration (Cmax) and time to reach Cmax (Tmax) were read directly from the experimental data. Statistical comparisons between pharmacokinetic parameters of the two products were analyzed using two-way ANOVA with *p* < 0.05 for statistical significance to assess the effect of origin of glucosamine, periods, sequence, subjects within sequence using IBM SPSS Statistics 25 (IBM Corporation, Armonk, NY, USA). The 90 percent confidence intervals (CIs) of the Biofermentation-derived glucosamine/Chitosan-derived glucosamine ratio of AUC_0–t_, AUC_0__–∞_, and Cmax using log transformed data were determined. The bioequivalence between the two different origins of glucosamine products would be accepted if the 90 percent CI of the log transformed AUC_0__–t_, AUC_0__–∞_, and Cmax of Biofermentation-derived glucosamine fell within 80–125% of the Chitosan-derived glucosamine [16].

## 3. Results

### 3.1. Subject Characteristics

This study was a double-blind, single-dose, fasting and randomized cross-over design recruiting 20 healthy young male and female subjects. Of the 20 subjects, 2 subjects were excluded from this study due to skin irritation from catheter use for 8 h irrespective of study supplementation. Therefore, 18 subjects completed both chitosan-derived formulation (R) treatment and biofermentation-derived formulation (T) treatment and we analyzed data for PK (Table 1).

### 3.2. Pharmacokinetic Data for Plasma Glucosamine in Subjects

Representative chromatograms of glucosamines two anomers are presented in Figure 1. Blank human plasma (Figure 1A) showed no endogenous interference at the retention times of glucosamine anomers. Human plasma spiked with glucosamine at the LLOQ level (Figure 1B), and a plasma sample obtained at 1 h after oral administration of glucosamine product (Figure 1C) showed separate peaks of the two anomers of glucosamine.

The endogenous glucosamine levels were below the LLOQ level in all subjects included. The mean plasma concentrations–time profile of glucosamine after an oral administration of two different glucosamine products to healthy volunteers are shown in Figure 2. Relative pharmacokinetic parameters of glucosamine and 90 percent CIs of Biofermentation-derived glucosamine/Chitosan-derived glucosamine ratios of log transformed AUC_0__–8h_, AUC_0__–∞_, and Cmax values are summarized in Table 2.

There were no statistically significant effects of sequence, period, origin of glucosamine on major pharmacokinetic parameters of glucosamine such as AUC_0__–8h_, AUC_0__–∞_, and Cmax. The AUC_0__–8h_ and AUC_0__–∞_ values of glucosamine following oral administration of chitosan-derived (reference, R) and biofermentation-derived (test, T) glucosamine formulations were within the bioequivalence criteria (90% CI of T/R ratios are within 0.8–1.25). However, 90 percent CI of T/R ratio of glucosamine Cmax values following the oral administration of the two products was not within the acceptable bioequivalence criteria.

## 4. Discussion

In this study, we successfully evaluated the pharmacokinetic profile for biofermentation-based glucosamine administrated in healthy subjects (Table 2). No adverse effects were observed. Not only did we successfully evaluate the pharmacokinetics parameters of biofermented glucosamine, our current data for the first time demonstrated that glucosamine from both chitosan and biofermentation source followed the similar PK parameters, i.e., AUC_0–8h_, AUC_0–__∞_, Cmax, Tmax and T1/2, (Table 2). Furthermore, as assessed by AUC_0–8h_ over AUC_0–__∞_ the relative bioavailability of test formula was (90.5%), vs. reference formula (92.7%).

The PK values of glucosamine from both sources were in consistent with those reported elsewhere. For example, Wu and co-authors reported a mean Cmax 944 ng/mL at 3.3 h, and mean AUC_0–__∞_, 3091 ng/mL/h, after 1500 mg glucosamine KCL dose was administrated by healthy subjects, with a mean half-life elimination of 1.5 h [17]. Using a glucosamine sulfate in solution dose, a mean Cmax of 1022 ng/mL was reached with a Tmax of 3.2 h, and mean value of AUC_0–4h_ and AUC_0–__∞_ obtained were 4034 and 4194 ng/h/mL, respectively [18]. In a study with all male subjects, slightly lower Cmax and AUC_0–__∞_ were reported [19]. It is known that many physical–chemical characteristics, formula format, gastrointestinal condition, food and other substances as well as gender and pharmacogenetic factors may influence the bioavailability of active compounds ingested as nutrients and therapeutic drugs.

Moreover, our current data provide, for the first time, a bioequivalence comparison between a chitosan glucosamine source and a biofermentation source available for use in dietary supplements. Though the chemical characteristics of chitosan-based and bio-fermentation-based glucosamine are identical, and safety of fermented glucosamine has been addressed (EFSA 2009), the bioequivalence was not known [20]. The confirmation of the bioequivalence of biofermented glucosamine will allow the confident switch from chitosan-based glucosamine to biofermented alternate while benefits and claims remain unchanged. Modern consumers prefer to limit animal-based dietary ingredient and are continuously adopting plant-based diet and dietary supplement options to fulfil their individual health needs and social responsibility [21,22]. Biofermented glucosamine provides a non-animal source alternative to consumers who are more conscious about the source of material used in their dietary supplement. We designed the current study to compare the pharmacokinetics to understand if biofermented glucosamine would follow the same pharmacokinetic patterns as that of chitosan-based glucosamine using FDA guidance on bioequivalence evaluation [14]. By comparing the % confidential interval of the AUC_0–8h_, AUC_0–__∞_ and Cmax between test candidate (biofermented glucosamine) and reference (chitosan-based glucosamine) were 93.4–121.1%, 91.4–117.0% and 89.2–134.2%, respectively.

Due to high intrasubject variation (ANOVA-CV of 35.1%), the Cmax of fermented and chitosan-derived glucosamine preparation did not meet the bioequivalence criteria falling just outside at 89.2−134.2%. Similar high inter-individual variation was reported elsewhere in other glucosamine pharmacokinetic studies [23,24]. In addition, sampling points near Tmax (1–2 h) might be too sparse to adequately determine the Cmax of each subject. Data analysis showed the geometric mean ratios of the two preparations were similar (geometric mean ratio of Cmax (T/R) was 1.09) but not statistically different. Moreover, bio-fermentation-derived glucosamine exhibits an equivalent exposure to glucosamine derived from chitosan. Therefore, it is reasonable to state that biofermentation-derived glucosamine could replace the chitosan derived one as a source of glucosamine ingredient in joint health dietary supplement products and would be a sustainable raw material source for glucosamine supplements.

In addition to the bioavailability and bioequivalence outcomes from our current study, the biofermented glucosamine was well-tolerated and apparently safe without causing any serious adverse effects in our current study, which was consistent with previously reported results (EFSA 2009) [20], confirming that biofermented glucosamine could be a safe alternative for dietary supplement use for those looking for a non-animal-base glucosamine option. The global prevalence of joint health concern is rising regardless of countries and demographic groups [25,26,27].

## 5. Conclusions

Our current finding is the first to outline the bioavailability of biofermentation-based glucosamine, indicating bioequivalence to the traditional chitosan-based glucosamine. This newly available non-animal sourced glucosamine is well-tolerated and safe after oral administration. Our current finding paves a smooth transition from chitosan-based glucosamine to a novel non-animal-based alternate option for dietary supplement development with the benefits to support joint health.

## Figures and Tables

**Figure 1 nutrients-14-03213-f001:**
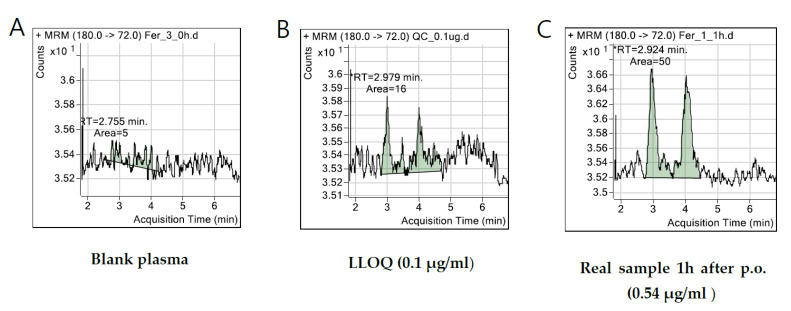
Representative MRM chromatograms of glucosamine in human plasma samples: (**A**) blank plasma; (**B**) blank plasma spiked with LLOQ level of glucosamine (0.1 μg/mL); (**C**) plasma samples at 1 h after an oral administration of 1500 mg of biofermentation-derived glucosamine sulfate 2KCl product to a healthy subject. Determined glucosamine concentration was 0.54 μg/mL. RT means retention time.

**Figure 2 nutrients-14-03213-f002:**
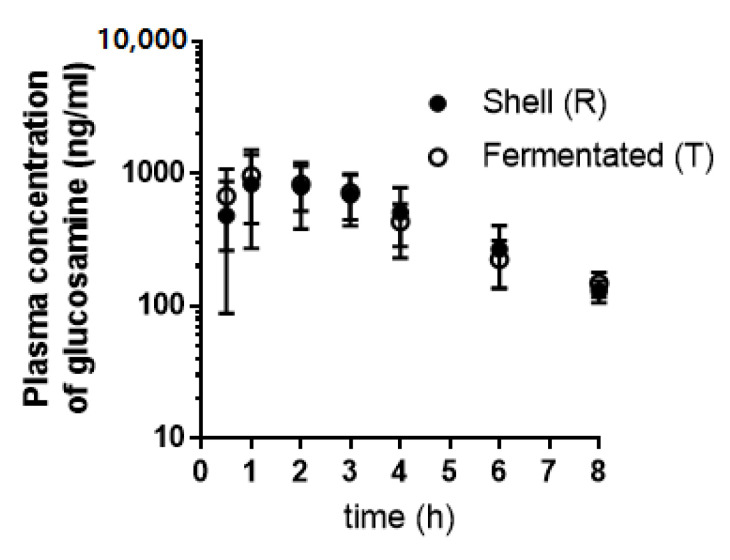
Mean plasma concertation-time profiles of glucosamine after an oral administration of 1500 mg of glucosamine sulfate 2KCl formulations to healthy volunteers. Shell (R), chitosan-derived formulation; and fermented (T), biofermentation-derived formulation.

**Table 1 nutrients-14-03213-t001:** General characteristics of enrolled subjects.

Parameters		Values
Gender	Male (*n*)	9
	Female (*n*)	9
Age (years)		24.3 ± 1.7
Weight (kg)		63.6 ± 8.7
Height (cm)		167.8 ± 7.6
Body mass index (kg/m^2^)		22.5 ± 2.3
Vital signs	Systolic blood pressure (mmHg)	132.5 ± 8.5
	Diastolic blood pressure (mmHg)	81.1 ± 9.7
Clinical laboratory	Hemoglobin (g/dL)	14.1 ± 1.4
	Hematocrit (%)	42.0 ± 3.9
	BUN (mg/dL)	12.5 ± 4.7
	Creatinine (mg/dL)	0.7 ± 0.1
	AST (units/L)	16.7 ± 3.7
	ALT (units/L)	13.8 ± 5.3
	ALP (units/L)	58.7 ± 13.7
	Total bilirubin (mg/dL)	0.49 ± 0.2

Blood clinical parameters were measured at 0 h; data presented as mean ± S.D.

**Table 2 nutrients-14-03213-t002:** Pharmacokinetic Parameters of Glucosamine.

Parameters	Chitosan-Derived Formulation (R)	Biofermentation-Derived Formulation (T)	90% CI of T/R Ratio
AUC_0–8 h_ (ng·h/mL)	3566 ± 1359	3709 ± 1217	(0.934–1.211) ^N.S.^
AUC_0–__∞_ (ng·h/mL)	3940 ± 1390	4000 ± 1240	(0.914–1.170) ^N.S.^
C_max_ (ng/mL)	1010 ± 530	1070 ± 510	(0.892–1.342) ^N.S.^
T_max_ (h)	2.0 ± 1.1	1.6 ± 1.0	ND
Terminal half-life (h)	2.15 ± 0.90	1.85 ± 0.61	ND

Data presents as Mean ± S.D.; N.S.: non-significant; ND: non-diagnosis.

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
