# Peer review of "Pharmacokinetic Comparison of Chitosan-Derived and Biofermentation-Derived Glucosamine in Nutritional Supplement for Bone Health"

_nutrients, 2022, doi:10.3390/nu14153213_

Round 1

Reviewer 1 Report

Nutrients-1787009

Pharmacokinetic comparison of chitosan-derived and biofer-2 mentation-derived glucosamine in nutritional supplement for 3 bone health

ABSTRACT

The lack of bioequivalence in terms of Cmax has to be mentioned.

INTRODUCTION

Lines 37-43: First part of the introduction highlights the use of glucosamine in combination with chondroitin sulfate, and it is well known that frequently glucosamine is used alone.

Lines 51-84: The Introduction is an absolute mess, with concepts repeated once and again. It needs to be revised in deep and rebuilt.

MATERIALS AND METHODS

Lines 90-91: Healthy male and female adults were recruited and assigned to take three capsules 90 per day.  It has to be clarified from the beginning if the 3 capsules are administered together or separately.

Sample size estimation is missing

Lines 92 and 95: 10 subjects were included in the first period and 9 in the second period. No explanation is provided for the subject missing.

The characteristics of bio-fermented glucosamine are not described.

Line 210: What the authors refer to with “There were no statistically significant effects of consequence”. It also appears in the abstract.

Lines 98- 99: the sampling time up to 8 hours looks too short considering the posology of glucosamine once a day (better sampling up to 24h).

Considering the Tmax for glucosamine in this and other studies is between 1-2 hours, the time points for sampling are not adequate to appropriately characterize the Cmax. Possibly this is the reason for the lack of bioequivalence found for this parameter. This should have been discussed in the discussion section.

Lines 117 onwards: the description of analytical method should be shortened considering the type of publication.

DISCUSSION

Sentences from the introduction are also repeated here.

The results of bioequivalence should be discussed as mentioned above.

Risks of biofermentation source should be discussed.

Reference 14 is missing. Other references in the paper have no relation with the text.

Reviewer 2 Report

Conflicts of interests, are five co-authors employees of the Access Business Group International, LLC, which also supported this study? This should be a typical example of conflicts of interests that needs to be reported.

Introduction:

Lines 68-84, the paragraph is basically the same as the above paragraph

Materials and Methods

Lines 90-96, need to revise to better describe the randomized crossover design. If it was a randomized crossover study, there should have one group not two groups of characteristics of subjects (Table 1).

Any analysis was done on the bio-fermented and chitosan-derived glucosamine capsules?

Table 1, at what time point blood clinical parameters were measured, 0 or 8h?

Discussion

Lines 240-241, what does this mean?

Round 2

Reviewer 1 Report

Section 3. Results

Subject characteristics

The description is confusing. To say that "9 subjects in the chitosan-derived formulation and 9 in the biofermentation-derived formulation completed the supplementation in the PK analysis" is incorrect. If the study follow a cross-over design, the eighteen subjects completed both treatment options. This is a misconception and I am concerned about it.

In addition, the wording "in the PK analysis" should be changed to "and analyzed for PK".

The wording "There was a general characteristics of subjects" is not correct.

The whole paragraph must be rewritten.

Author Response

Section 3. Results

Subject characteristics

The description is confusing. To say that "9 subjects in the chitosan-derived formulation and 9 in the biofermentation-derived formulation completed the supplementation in the PK analysis" is incorrect. If the study follow a cross-over design, the eighteen subjects completed both treatment options. This is a misconception and I am concerned about it.

In addition, the wording "in the PK analysis" should be changed to "and analyzed for PK".

The wording "There was a general characteristics of subjects" is not correct.

The whole paragraph must be rewritten.

: We appreciated a valuable review of the reviewer and apologized the confusing part in the paragraph.

Therefore we provided the whole new paragraph in revised manuscript as reviewer requested. Please see section 3. Results.

Reviewer 2 Report

The revisions is acceptable.

Round 3

Reviewer 1 Report

The information contained in the first paragraph of section 3 has been improved but it is necessary to review the wording. This also applies to the English language and style throughout the manuscript.
